# WHAT IS THE CHANCE OF BEING SO UNFAIR?

## ABSTRACT

Fairness has often been seen as an ethical concern that needs to be considered at some cost on the utility. In contrast, in this work, we formulate fairness, and especially fairness in ranking, as a way to avoid unjust biases and provide a more accurate ranking that results in improvement on the actual unbiased utility. With this in mind, we design a fairness measure that, instead of blindly forcing some approximate equality constraint, checks if the outcome is plausible in a just world. Our fairness measure asks a simple and fundamental statistical question: "What is the chance of observing this outcome in an unbiased world?". If the chance is high enough, the outcome is fair. We provide a dynamic programming algorithm that, given a ranking calculates our fairness measure. Secondly, given a sequence of potentially biased scores, along with the sensitive feature, we provide a fair ranking algorithm based on our fairness measure. Finally, we run some experiments to understand the behavior of our ranking algorithm against other fundamental algorithms.

## 1 INTRODUCTION

In the past decade, fairness has become a key concept in machine learning and automated decision making. Specifically, in recommendation systems and hiring platforms, fairness means that ranking mechanisms should be unbiased and not discriminate based on demographic characteristics or other protected attributes.

The group of individuals in a ranking task is called candidates who may have sensitive attributes. The algorithmic methods that address fairness differ in the representation of candidates, the type of bias, mitigation objectives, and mitigation methods such as worldviews Zehlike et al. (2021). In response to worldviews, Friedler et al. Friedler et al. (2021) highlight the need to understand the difference between our belief about fairness and the mathematical definition of fairness. They present two views that represent the two ends of the spectrum: WYSIWYG ("what you see is what you get") and WAE ("we are all equal"). WYSIWYG assumes that what we see is nearly the same as the real properties, with just a $\epsilon$ distortion. WAE assumes that biased observations cause differences in utility distributions among the candidates.

In this work, we introduce a stochastic variant of WAE, that we refer to as *Stochastic-WAE*. Based on stochastic-WAE, we provide a fairness measure that poses a fundamental statistical question: *What is the likelihood of observing this outcome in an unbiased scenario?*. If this likelihood is high enough, we consider it fair. We present Stochastic-WAE that captures randomness while keeping it independent of sensitive data. It recognizes that probability distributions for different groups, such as females and non-females, should be the same, despite potential score gaps in specific subsets.

Given a ranking, we provide a dynamic programming algorithm that answers the above question and calculates our fairness measure. This can be used on top of other ranking algorithms to measure their fairness. Next, we design a ranking algorithm that respects our fairness measure. Specifically, we design an algorithm that, given a measure $\delta$ and given a sequence of potentially biased scores, along with the sensitive feature, provides a fair ranking with maximum possible utility such that its fairness measure is at most $\delta$.

## 1.1 PROBLEM SETTING

Let $\Omega$ be the set of all possible candidates that originate from a society that is originally faced with bias. For simplicity of presentation, we focus on a single binary group with a score that includes an unknown preexisting bias against women. Let

$$id(\omega) = \begin{cases} 1 & \text{if } \omega \text{ is a woman,} \\ 0 & \text{otherwise,} \end{cases}$$

where $\omega \in \Omega$. We receive the set of candidates $\mathcal{C} = (\omega_1, \dots, \omega_n)$ and their corresponding biased scores $(\hat{y_1}, \hat{y_2}, \dots, \hat{y_n})$. We define the following quantity which represents the *minority proportion*

$$p = \frac{\sum_{j=1}^n id(\omega_j)}{n}.$$

A permutation over the candidate sample $\mathcal{C}$ is called a *ranking*. Let $\tau : \{1, \dots, n\} \to \mathcal{C}$ be the observed-score-based ranking on $\mathcal{C}$, i.e., $\hat{y}_{\tau(1)} \geq \dots \geq \hat{y}_{\tau(n)}$, in which $\tau(i)$ is the candidate at rank $i$. The utility of $\tau$ based on DCG approach is defined as follows Zehlike et al. (2022):

$$U(\tau) = \sum_{i=1}^n \frac{\hat{y}_{\tau(i)}}{\log_2(i+1)}. \tag{1}$$

In the context of ranking algorithms, the worldview of "We are all equal (WAE)" implies that individuals with similar qualities should have an equal chance of being ranked similarly Zehlike et al. (2022).

In this work, we adopt a statistical approach to similarity, and hence, we assume that the unbiased scores for men and women, in which discrimination based on gender is absent, are taken from the same unknown probability distribution. Let $y_i$ be the $i$-th candidate's unbiased score which is taken from an unknown probability distribution $\mathcal{P}_Y$ where a random variable $Y : \Omega \to \mathbb{R}$ represents the unbiased score for a given candidate. Based on the statistical WAE worldview, the value of $y_i$ is independent of the group to which the candidate $\omega_i$ belongs. Therefore, $(y_1, \dots, y_n)$ is an independent and identically distributed (i.i.d.) random sequence. Let the permutation $\sigma$ be an ordering of $y_1, \dots, y_n$ s.t. $y_{\sigma(1)} \geq \dots \geq y_{\sigma(n)}$. The permutation $\sigma$ is an *unbiased ranking* for $\mathcal{C}$.

Since the unbiased scores are taken the same distribution, we have $\mathbb{E}\left[Y(\omega) \mid id(\omega) = 1\right] = \mathbb{E}\left[Y(\omega) \mid id(\omega) = 0\right]$, and consequently, intuitively, the arrangement of women and men in cut-off points of the ranking should not be statistically rare. Next we formalize this notion.

## 1.2 DEFINING FAIRNESS IN RANKING VIA STATISTICAL-WAE WORLDVIEW

For a ranking $\tau : \{1, \dots, n\} \to \mathcal{C}$, let $\mathcal{F}_\tau = \{S_1^\tau, \dots, S_n^\tau\}$ where $S_i^\tau = \{\tau(1), \dots, \tau(i)\}$. We call $S_i^\tau$ the *i'th partial set corresponding to* $\tau$. Hence, $S_1^\tau \subset S_2^\tau \subset \dots \subset S_n^\tau$. Let $X_i^\tau$ be the number of women in $S_i^\tau$. Sort $Y(\omega_1), Y(\omega_2), \dots, Y(\omega_n)$ to obtain an unbiased ranking $\sigma$ such that $Y(\omega_{\sigma(1)}) \geq Y(\omega_{\sigma(2)}) \geq \dots \geq Y(\omega_{\sigma(n)})$.

Similarly, $X_i^\sigma$ is the number of women in the $i$'th partial set $S_i^\sigma$. Since $Y$ is a random variable, one can see that $\sigma$ is a random permutation and so $X_i^\sigma$ is a random variable. For the sake of simplicity, we will denote $X_i^\sigma$ by $X_i$ in the rest.

**Definition 1.** *For a ranking* $\tau : \{1, \dots, n\} \to \mathcal{C}$, *the partial set* $S_i^\tau$ *is said to be "$\delta$-rare" if and only if the following inequality holds:*

$$Pr[X_i \leq X_i^\tau] < \delta.$$

*Moreover, we say* $X_i^\tau$ *is in the $\delta$-tail of* $\mathcal{P}_{X_i}$.

Now, we want to formalize the notion of fairness concerning the statistical WAE worldview precisely.

**Definition 2.** *A ranking* $\tau : \{1, \dots, n\} \to \mathcal{C}$ *is called $\delta$-fair if and only if none of the members of the* $\mathcal{F}_\tau$ *are $\delta$-rare.*

This definition explicitly says that none of the partial sets associated with a fair ranking is in the $\delta$-tail of $\mathcal{P}_{X_i}$. In other words, a ranking is $\delta$-fair, if the occurrence probability of the least probable partial set is not lower than $\delta$.

## 2 OTHER RELATED WORKS

One simple approach to define fairness is that the proportion of women among the top batches of candidates must be close to the minority proportion. Yang et al. Yang and Stoyanovich (2017) propose several quantifiers for this notion such as *Normalized discounted difference* and *Normalized discounted ratio* in which they calculated the sum of discounted differences between the portion of women and men in some cut-offs of the ranking.

Kleinberg and Raghavan introduced another approach in this framework Kleinberg and Raghavan (2018) in which they assumed the candidates are partitioned into two groups (protected and privileged) and each candidate has an unknown unbiased score which is called *potential*. Moreover, same as what we assumed for the unbiased scores, they studied the case that the potentials for both groups come from a power law probability distribution. To maintain fairness, they propose to discredit the observed scores for the privileged group by downgrading them by a factor. This work enforces some assumptions on the probability distribution of the potential function. In comparison, we do not require any assumptions on the distribution. In both works, it seems likely that a member of the privileged group faces unfair discrimination since the main concern is to give some artificial benefits to the protected group to maintain the desired diversity in the outcome.

In the context of mitigation objectives, in addition to worldviews that we mentioned before, there are two other main concepts, namely, *Equality Opportunity*, and *Intersectional discrimination* Zehlike et al. (2021). Equality of Opportunity (EO) is a philosophical idea which intends to eliminate unfair barriers so that everyone has a fair chance to reach good positions in life Friedler et al. (2021); Hardt et al. (2016;?); Khan et al. (2021); Kleinberg and Raghavan (2018); Dworkin (1981); Roemer (2002); Zehlike et al. (2020); Arneson (2018); Khan et al. (2021). Heidari et al. Heidari et al. (2019) use the EO framework to figure out how a person's outcome is influenced by two main factors: circumstance and effort. Circumstances include factors that are not the individual's acts, such as gender, race, and the family they were born into. The effort includes factors such as the individual's decisions and acts that can justify differences. There are different ideas about EO, such as, which factors to consider and how to model the relationship between circumstance and effort.

Information Access Systems (IAS) rank and display content based on perceived merit, with content producers increasingly recognized as important stakeholders Joachims (2002). These interests can be assessed individually or by group characteristics like gender or race. One of measures for promoting fairness in rankings is pairwise accuracy Kuhlman et al. (2019); Fabris et al. (2023a;b).

Intersectional Discrimination states that candidates may belong to multiple protected groups at the same time, like being both African-American and Female Crenshaw (1997); Makkonen (2002); Schumacher et al. (2024) and seeks fairness for both simultaneously Collins (2022); Noble (2018); Shields (2008); Yang et al. (2019). In the context of ranking, if fairness means having a fair share in the top positions, it might be possible to have fairness for each gender group (men and women) and each racial group (caucasian and non-caucasian) separately. However, there could still be a problem if you look at a group that's both Black and women, like Black women. They might not be well-represented, even if each gender and racial group seems okay on its own Collins (2022); Noble (2018); Shields (2008); Yang et al. (2019).

Score-based and supervised learning-based ranking methods employ distinct strategies to tackle bias issues Zehlike et al. (2021); Hajian et al. (2016). In score-based ranking Yang and Stoyanovich (2017); Yang et al. (2020; 2019); Stoyanovich et al. (2018); Kleinberg and Raghavan (2018); Celis et al. (2017; 2020); Asudeh et al. (2019), three key approaches are utilized to address bias. The first involves intervention in the score distribution, to reduce inequality inequalities in candidate scores. The second approach is scoring function intervention which includes modifying how the scoring scoring process operates. The third aspect focuses on intervening in the ranked outcome to ensure the final ranked list is fair for everyone.

In supervised learning Biega et al. (2018); Beutel et al. (2019); Geyik et al. (2019); Lahoti et al. (2019); Singh and Joachims (2018; 2019); Zehlike et al. (2017; 2020; 2022), bias mitigation methods are categorized into three main groups: pre-processing, in-processing, and post-processing. Pre-processing methods focus on rectifying bias in the training data. In-processing methods aim to train models that inherently lack bias. Post-processing methods come into play after generating rankings,

re-evaluating, and adjusting the ranking outcomes based on specific fairness criteria Zehlike et al. (2021).

The objective function of the model aims to find a balance among three components: application utility (i.e. classifier accuracy), group fairness, and individual fairness. However, this can make the learning process challenging as it involves managing multiple aspects that may not align easily Lahoti et al. (2019). Zehlike et al. Zehlike et al. (2020) proposed an algorithm by utilizing the optimal transport theory to optimize decision-maker utility within the constraints of fairness. In another work Zehlike et al. Zehlike et al. (2021) explored various several approaches that employ distinct interpretations of utility, and we will clarify their formulations as needed. They also describe various interpretations of utility. In score-based ranking, the simplest method of determining utility is the sum of the scores of its elements disregarding candidate positions. Another approach involves incorporating discounts based on position. This is rooted in the observation that placing high-quality items at the top of the ranked list is more crucial, as these items are more likely to capture the attention of the consumer of the ranking. An alternative approach involves measuring the utility achieved by candidates from a specific demographic group.

## 3 Ranking Algorithms

In this section, we provide an algorithm to measure the fairness in ranking based on our fairness criteria. Next, we design an algorithm that fairly ranks a given candidate set.

### 3.1 An Algorithm to Measure Ranking Fairness

In algorithm 1, as we assumed the unbiased scores of all candidates come from the distribution $\mathcal{P}_Y$, at each step of a ranking (consider the process as a step-by-step procedure that puts the candidates in their place respectively from the first to the $n$th place), the probability of the next candidate to be a woman is the proportion of unranked women to the total number of remaining candidates. We prove the following theorem.

**Theorem 1.** *For a given candidate sample $\mathcal{C} = (\omega_1, \cdots, \omega_n)$, let $n_1$ and $n_2$ be the total number of women and men respectively. Let a tuple $(i, m)$ represent the event of seeing $m$ men in the first $i$ candidates of a fair ranking. Then, by statistical WAE worldview, the following equation holds*

$$Pr\left[(i, m)\right] = \left(\frac{n_2 - (m-1)}{n - (i-1)}\right) Pr\left[(i-1, m-1)\right]$$
$$+ \left(\frac{n_1 - (i-1-m)}{n - (i-1)}\right) Pr\left[(i-1, m)\right].$$

This theorem allows us to calculate the probability of a partial set using the probabilities of the previous step's partial sets. This enables us to develop a dynamic programming algorithm to calculate the partial set probabilities which is represented in Algorithm 1.

By Theorem 1 the probability of the event that at most $k$ women are among the first $i$ candidates in an unbiased environment, $Pr[X_i \leq k]$, is stored in $P[i, i - k]$ (defined in line 9 of Algorithm 1). Hence, we can verify the $\delta$-fairness of a given ranking using Algorithm 2.

### 3.2 Obtaining Fair Ranking With Highest Utility

The main goal of Algorithm 3 is to find a fair ranking that has the maximum utility among all possible fair rankings. In order to do so, we follow a sequence of greedy operations and use dynamic programming to choose the best (highest utility) ranking at each step. The following theorem allows us to construct the required ranking permutation inductively, as the algorithm 3 does.

**Theorem 2.** *For a positive real number $\delta$ and a candidate set $\mathcal{C}$, Algorithm 3 outputs a $\delta$-fair ranking that has the highest utility.*

---

**Algorithm 1** Probabilities Of Partial Sets

---

1: **Input** Dataset of candidates $\mathcal{C}$.
2: Let $n, n_1$ and $n_2$ be the total number of candidates, women, and men respectively.
3: Let $Q$ be a $n \times n_2$ matrix in which the $Q[i, m]$ corresponds to the probability of the event that $m$ men occur in the first $i$ candidates of an unbiased ranking.
4: Initialize $Q$:

$$Q[i, m] = \begin{cases} 0 & : \ i < m \\ \frac{n_1}{n} & : \ (i, m) = (1, 0) \\ \frac{n_2}{n} & : \ (i, m) = (1, 1) \end{cases}$$

5: **for** $i = 1, 2, \ldots n$ **do**
6:     **for** $m = 1, 2, \ldots, \min(i, n_2)$ **do**

$$Q[i, m] = \left( \frac{n_2 - (m - 1)}{n - (i - 1)} \right) Q[i - 1, m - 1]$$
$$+ \left( \frac{n_1 - (i - 1 - m)}{n - (i - 1)} \right) Q[i - 1, m]$$

7:     **end for**
8: **end for**
9: Let $P$ be a $n \times n_2$ matrix in which the $P[i, m]$ corresponds to the probability of the event that at least $m$ men are among the first $i$ candidates of an unbiased ranking.
10: Initialize $P$:

$$P[i, m] = \begin{cases} 0 & : \ i < m \\ 1 & : \ m = 0 \\ \frac{n_2}{n} & : \ (i, m) = (1, 1) \end{cases}$$

11: **for** $i = 1, 2, \ldots n$ **do**
12:     **for** $m = 1, 2, \ldots, \min(i, n_2)$ **do**

$$P[i, m] = \sum_{j=m}^{i} Q[i, j]$$

13:     **end for**
14: **end for**
15: **return** $P$

---

**Algorithm 2** VerifyFairnessByPartialSets

---

1: **Input** Dataset of candidates $\mathcal{C}$, a permutation (ranking) function $\tau$, a real positive number $\delta$
2: Let $n$ be the total number of candidates.
3: $P \leftarrow$ Probabilities Of Partial Sets$(\mathcal{C})$
4: **for** $1 \leq i \leq n$ **do**:
5: $m_i =$ number of men in $\{\tau(1), \ldots, \tau(i)\}$
6:     **if** $P[i, m_i] < \delta$ **then** report **unfair** and terminate.
7:     **end if**
8: **end for**
9: Report **fair**.

---

## 4 EXPERIMENTAL RESULTS

In this section, we report the experimental results in which we compared the average *true utility* of several algorithms on several synthetic data sets. Synthetic datasets are artificially created datasets that imitate the properties and structure of real-world data through a clear and understandable process. By true utility we mean the value of the utility function (that is introduced in (1)) on the unbiased scores which in reality we are not aware of, but since we are using synthetic datasets, we

---

**Algorithm 3** FindTheBestRankingByPsets

---

1: **Input** candidate set $\mathcal{C}$, observed scores of candidates $\hat{Y}$, a real positive number $\delta$
2: Let $n, n_1$ and $n_2$ be the total number of candidates, women, and men respectively.
3: Let $Y_w$ and $Y_m$ be the sorted scores of women and men, respectively.
4: $P \leftarrow$ CalculateProbabilitiesOfPartialSets($\mathcal{C}$)             ▷ Algorithm 1
5: Let $U$ be an $n \times n_2$ matrix that stores the highest utility that can be obtained by a fair ranking of $i$ candidates with $m$ men among them in $U[i, m]$.
6: Let $R$ be a table with lists as entries that store the corresponding ranking of $U[i, m]$.
7: Initialize $U$ as follows:

$$U[1, 0] = \begin{cases} Y_w[0] & \text{If } P[1, 0] > \delta \\ -\infty & \text{O.W} \end{cases}$$

$$U[1, 1] = \begin{cases} Y_m[0] & \text{If } P[1, 1] > \delta \\ -\infty & \text{O.W} \end{cases}$$

8: **for** $i = 2, \ldots, n$ **do**
9:      **for** $m = 0, \ldots, \min(i, n_2)$ **do**
10:          **if** $P[i, m] < \delta$ **then** $U[i, m] = -\infty$
11:              break
12:          **end if**
13:          **if** $i - m - 1 < n_1$ **then**

$$u_1 = \frac{Y_w[i - m - 1]}{\log_2(i + 1)} + U[i - 1, m]$$

14:          **end if**
15:          **if** m ¿ 0 **then**

$$u_2 = \frac{Y_m[m - 1]}{\log_2(i + 1)} + U[i - 1, m - 1]$$

16:          **end if**
17:          Handle the extreme cases of $i - 1 - m = n_1$ and $m = 0$.
18:          $U[i, m] = \max(u_1, u_2)$
19:          Update $R$
20:      **end for**
21: **end for**
22: Let $\pi = R[n, n_2]$.
23: **Output** $\pi$

---

can assume that the unbiased scores are provided initially. Each data set consists of a set of candidates which are grouped by their gender and a set of unbiased scores for all of them which comes from a distribution independent of their gender. As the literature implies, we assume the male candidates are the privileged ones so we set the observed scores of the male candidates the same as their unbiased scores. The observed scores of women candidates are obtained by their unbiased scores decremented by a random bias. We assume the unbiased scores come from a normal distribution and without loss of generality[1], we set the average to be 15. We report the results for two different standard deviations 5 and 10. The distribution of bias may vary, but here because of the paper size limit, we just study two cases of Normal and Uniform and Exponential distribution. For the case that the bias comes from Normal or Exponential distribution, we report the results for the bias average range of 0 to 5 and for the Normal case specifically, we report the outcome for three different standard deviations 0.5, 1, and 2 which seem more realistic in practice. In the body of the paper, we study the cases where the number of male and female candidates are equal. In the appendix, we provide the experiments where the portion of men and women are not equal.

---

[1]Because by the linearity of expectation, if we add a constant value to all of the scores, the mean of the scores would be shifted by the same value. Moreover, this constant shift does not change the order of the candidates and the utility function as defined in (1) would be shifted by a function of that constant value.

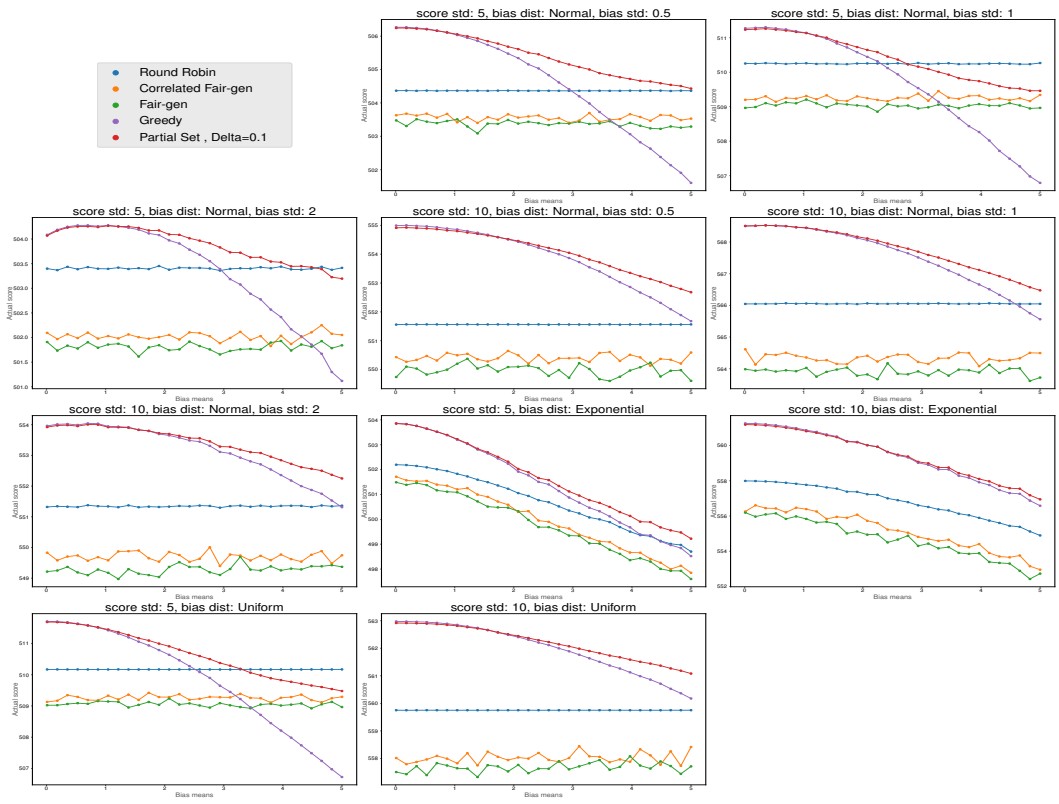

Figure 1: Utility of Algorithms Over Unbiased Scores, 50 women - 50 men

We implemented Algorithm 3 (which is called *Partial Set* in our graphs) as well as some other algorithms motivated by the literature. Here we re-introduce some previously studied algorithms that will be used in our experiments.

**Round Robin:** This is the most trivial approach for satisfying fairness criteria. If the portion of men to women is $\beta$, we simply put the best-unranked woman after each $\beta$ men. For the sake of simplicity, we suppose $\beta$ is 1, $\frac{1}{3}$ and 3.

**Correlated Fair Gen:** Yang et al. Yang and Stoyanovich (2017) propose an algorithm (called Ranking Generator) which randomly ensures that the number of protected candidates does not fall far below the minority proportion $p$. For each step of the Ranking generator algorithm, a Bernoulli experiment with the success probability $1 - p$ is done and if the experiment succeeds, we put a man in that place.

**Correlated Fair Gen:** In the correlated approach, called *Correlated Fair Gen*, we update the minority proportion $p$ in each step and place the candidate using a Bernoulli experiment as in the above algorithm. This modified version is represented here just to enrich our experiments.

Due to the page limit, we just report the experiments of the cases in which the population of men and the population of women are equal. We note that, we do not observe a huge change in the behavior of the algorithms when we change the portion of men and women.

The main statement that we want to conclude from these experiments is that the Partial Set algorithm, which is in some sense more moderate than Greedy and Round Robin, *almost* every time can do better than both of them on unbiased scores. Because the Partial Set algorithm cares about fairness and utility at the same time and is a mix of Greedy and Round Robin reasonably. In the following experiments, the comparisons clarify when our algorithm does and when it does not better than the other two.

As shown in Figure 3.2, in all of the experiments, Partial Set and Greedy algorithms are almost the same when the bias average is low. And when the bias average increases, Greedy goes down faster than any other algorithm and if the bias average is not higher than a large value, the Partial Set algorithm has the highest utility among them all. But when the bias average exceeds a certain threshold, the Round Robin wins and it makes sense.

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
