# OpenReview forum: "What is the chance of being so unfair?"
_ICLR.cc/2025/Workshop/BuildingTrust — Submitted to BuildingTrust_

### Official Review · Reviewer_oBre · 2025-02-24
**Review of “What Is the Chance of Being So Unfair?”**

**Rating:** 2
**Confidence:** 3

**Review:**

**Paper summary:**

In this paper, the authors propose a group fairness metric for ranking applications, based on a statistical difference between probability distributions of utility across different groups. Furthermore, they propose a dynamic programming algorithm to compute this metric, develop a “fair ranking” algorithm based on the metric, and conduct experiments to compare it against other “fair ranking” algorithms.

**Reasons to accept:**

1.	It is interesting to approach the problem of group fairness as a statistical problem with each group represented as a probability distribution.

**Reasons to reject:**

1.	According to Figure 1, where the authors’ method does perform better than other methods, the improvement is extremely marginal, which heavily casts into doubt the novelty and utility of their approach compared to existing metrics.
2.	There is a lack of motivation of why the author’s approach should be used over state-of-the-art ones [1], or the specific limitations of other metrics / algorithms that the author’s approach can address. The authors mention certain other methods are “blindly forcing some approximate equality constraint” (lines 14-15), but do not mention why that this existing approach is problematic or limited in any way.
3.	There is a lack of theoretical support for the superiority or effectiveness of the proposed metric and algorithms. Theorems 1 and 2 have no associated proof, and the components of the mathematical formulations (e.g. what the fractions in Theorem 1 represent) are not explained in adequate detail.
4.	The proposed algorithm is limited by the strict requirement for all partial sets in 𝜏 to be δ-rare. For many practical applications, such a ranking, while possible, may only be able to produce a very suboptimal utility. The authors do not consider if and how this assumption can be relaxed under certain conditions.
5.	The authors don’t experiment with real-world datasets, only synthetic ones. This is problematic because real-world data is often more noisy or complex and does not strictly obey a mathematical distribution. In this way, the authors do not show how the algorithm can perform in practice for various applications.
6.	There is no discussion on the real-world utility of proposed metric and algorithm, limitations, or future work.
7.	The authors mention an Appendix (line 319), but it is not present in the final submission.

**Suggestions for authors:**

1.	In much of the paper, the approach is framed using a single example with women and non-women. I would suggest framing it in a more generalizable way, not simply tailoring to a single example.
2.	I would recommend against using the word “minority” (line 62), as the fairness problem can also apply to sets of groups where there is no clear or fixed minority. I would recommend using “protected class” or “sensitive class” instead.
3.	I would recommend against using “unbiased” (e.g. line 75) – it’s a bit misleading because (to me) it assumes that these scores exist in practice rather than hypothetically. I recommend using something like “theoretically unbiased” instead.

[1] M. Zehlike et al. Fairness in ranking: A Survey. 2021.

---

### Official Review · Reviewer_x28q · 2025-02-27
**Out of Scope of the Workshop**

**Rating:** 2
**Confidence:** 4

**Review:**

My review will be very brief, as I only broadly perused the submission. This is because I found that the paper is not at all in scope of the workshop.

The paper proposes a new fairness measure in the context of ranking, and then argues for why this fairness metric is the right way to discuss fairness in ranking. Finally, they propose an algorithm that can turn an 'unfair' ranking into a 'fair' ranking under this metric.

The problem setting in the paper does not incorporate LLMs at any point. The setup is ranking, solved using traditional techniques. The dataset used is tabular. In fact, in a brute force attempt to find some connection, I even 'searched' for the terms 'LLM' or 'language' in the entire paper, finding not even a single occurrence.

This might or might not be a good paper. The review is not a comment on the quality of the paper. However, the paper is completely out of scope of the workshop, and I don't believe the reviewers should be expected to have any further comments about a paper this far away from the actual scope of the workshop.

---

### Decision · Program_Chairs · 2025-03-04

**Decision:**

Reject

**Comment:**

This paper introduces a new fairness metric for ranking and proposes an algorithm to transform unfair rankings into fair ones based on this metric. However, it is deemed entirely out of scope for the workshop, as it does not involve LLMs or language-based models.